# Longitudinal imaging of *Caenorhabditis elegans* in a microfabricated device reveals variation in behavioral decline during aging

Matthew A Churgin[1], Sang-Kyu Jung[1], Chih-Chieh Yu[1], Xiangmei Chen[1], David M Raizen[2], Christopher Fang-Yen[1,3]*

[1]Department of Bioengineering, School of Engineering and Applied Sciences, University of Pennsylvania, Philadelphia, United States; [2]Department of Neurology, Perelman School of Medicine, University of Pennsylvania, Philadelphia, United States; [3]Department of Neuroscience, Perelman School of Medicine, University of Pennsylvania, Philadelphia, United States

**Abstract** The roundworm *C. elegans* is a mainstay of aging research due to its short lifespan and easily manipulable genetics. Current, widely used methods for long-term measurement of *C. elegans* are limited by low throughput and the difficulty of performing longitudinal monitoring of aging phenotypes. Here we describe the WorMotel, a microfabricated device for long-term cultivation and automated longitudinal imaging of large numbers of *C. elegans* confined to individual wells. Using the WorMotel, we find that short-lived and long-lived strains exhibit patterns of behavioral decline that do not temporally scale between individuals or populations, but rather resemble the shortest and longest lived individuals in a wild type population. We also find that behavioral trajectories of worms subject to oxidative stress resemble trajectories observed during aging. Our method is a powerful and scalable tool for analysis of *C. elegans* behavior and aging.

*For correspondence: cfangyen@gmail.com

**Competing interests:** The authors declare that no competing interests exist.

## Introduction

Aging consists of gradual changes in an adult organism that cause a reduction of function and an increase in mortality. Studies of model organisms such as the roundworm *C. elegans* have identified highly conserved processes and pathways which influence aging, including dietary restriction (*Lakowski and Hekimi, 1998*; *Greer and Brunet, 2009*), insulin/insulin-like signaling (*Kenyon et al., 1993*), and the cytoprotective DAF-16/FOXO pathway (*Greer and Brunet, 2007*; *Eijkelenboom and Burgering, 2013*; *Ogg et al., 1997*).

Current, widely used methods for long-term measurement of *C. elegans* aging are based on manual inspection of worm survival on agar plates. These methods are robust and technically simple, but have a number of limitations. They are labor intensive, low in throughput, and are largely focused on lifespan at the population level without access to information about health or behavior during individual life trajectories.

Previous efforts have aimed to automate *C. elegans* survival assays. One method produces high resolution survival curves by monitoring large populations of animals on standard agar plates using flatbed scanners (*Stroustrup et al., 2013*). However, this method monitors only lifespan and is not designed to track individual animals over their entire lifetime. Therefore, this system is limited in its ability to study aging in individual animals.

**eLife digest** Aging affects almost all living things, yet little is known about the biological changes that occur as we get older. Scientists often study aging in the microscopic roundworm *Caenorhabditis elegans* because it reproduces quickly and its lifespan is short (about 2–3 weeks on average). To date, investigations have helped to reveal genes that affect overall lifespan. However, it is not known how much these genes also affect the animal's healthy lifespan or "healthspan", that is to say, the length of time before advancing age begins to negatively affect health.

Until now, studies with worms have often been limited because measuring health and aging required time-consuming and difficult manual experiments. This also meant that worms were studied together as groups, rather than as individuals, providing a simplified picture of what was going on. An automated system in which many single worms can be analyzed and assessed would provide a much more detailed view of the effects of aging on health.

Churgin et al. have now developed a device called the WorMotel to allow simultaneous automated examination of 240 worms throughout their entire adult lifespan. The WorMotel is a rectangular slab of clear silicone rubber with small wells in it. A single worm is confined in each well with a source of bacteria for food, and a camera is used to track and monitor each worm's behavior over time. This device confirmed that worms move more slowly as they get older, which was taken to be a measurement of the worms' declining health. Worms that lived the longest declined over the first few days and then had a long plateau of very low activity before eventually dying. Short-lived worms became slower and died fairly promptly.

Churgin et al. also showed that the worms with mutations that increase lifespan declined in a similar way to the longest-lived normal worms, and that mutants with shorter lifespans declined like the shortest-lived normal worms. Also, normal worms that had been exposed to a chemical called paraquat – which stresses the worm's cells and shortens the worm's lifespans to a few days – slowed down in a similar manner as aging worms, suggesting that the stress is similar to the aging process.

Tools like the WorMotel can improve our understanding of the links between lifespan and healthspan. The tool is designed to be versatile and can be used with standard imaging systems and automated tools, meaning it can be scaled up to deal with tens of thousands of worms at once. Churgin et al. are now using the WorMotel to find other genes that influence healthspan and understand how they contribute to deteriorating health as animals age. Aging affects us all and learning more about healthspan could lead to drugs or interventions to help more people to live healthily for longer.

Another method (*Zhang et al., 2016*) used small hydrogel compartments between a glass slide and a PDMS membrane to perform long-term longitudinal monitoring of *C. elegans* aging. However, the device has a number of limitations: it is not easily scalable to large numbers of animals and does not lend itself to screening experiments. The hydrogel device requires complex image analysis software and prevents access to animals during the experiment, limiting the additional phenotypes that can be assayed in tandem. Also, the hydrogel device requires sterile mutations to be crossed into all strains tested, and all experiments must be performed at the restrictive temperature of 25° C, limiting the prospects for genetic screening.

Here, we describe a device we call the WorMotel (WM), which is capable of longitudinally tracking behavior of up to 240 uniquely identified animals of any genotype per device for over 60 days. The WM consists of an array of individual wells, each of which is filled with standard agar media, bacterial food, and a single worm, enabling long-term cultivation and imaging of hundreds of uniquely identified animals. By conforming to the ANSI standard microplate format, our method leverages existing scalable automation technology including worm sorters, robotic plate handlers, and chemical library screening tools. We apply our method to quantifying inter-individual and inter-strain differences in behavioral decline during aging and stress, as well as in understanding the relationship between behavior and lifespan.

## Results

### A scalable platform for long-term imaging of worm behavior, development, and lifespan

Conventional 96-well or 384-well microplates are not well suited for worm imaging and cultivation on agar media due to three problems. First, the vertical walls of each well make it difficult to image the worms when they are close to the edge of the wells (*Figure 1—figure supplement 1*). Second, worms tend to crawl between the agar and well edges, again making them difficult to image clearly. Third, under humid conditions, animals can climb over the walls of the wells, mixing with other worms.

We designed the WorMotel to address these limitations. Each WorMotel consists of a transparent polydimethylsiloxane (PDMS) substrate containing a rectangular array of up to 240 wells, produced by molding from an acrylic photopolymer 3D-printed master (*Shepherd et al., 2011*) (*Figure 1*, *Figure 1—figure supplement 2*). The well geometry is optimized for worm cultivation and imaging of a single worm per well. A rounded concave well geometry (*Figure 1a*) provides a clear view of the animal at all positions on the agar surface (*Yu et al., 2014*), and also inhibits worms from burrowing under the agar (*Figure 1—figure supplement 3*). A network of narrow moats containing a copper sulfate solution surrounding the wells prevents animals from escaping from their wells (*Figure 1b*).

Each well is filled with approximately 15 µL of NGM agar, followed by a suspension of bacteria added as food. For lifespan measurements, enough bacteria is added initially to sustain worms throughout their lives, and the animals are left essentially undisturbed for the remainder of the experiment. A single worm is added manually to each well.

The WorMotel is sealed inside a polystyrene dish (Nunc Omnitray, ThermoFisher Scientific) and imaged under dark field illumination with a CMOS camera (Imaging Source, Charlotte, NC) where it remains for the experiment's duration (*Figure 1e*) (*Churgin and Fang-Yen, 2015*). If desired, however, the device may be removed periodically for manual inspection or other longitudinal assays. The pixel resolution for a field of view containing all 240 wells is 36 µm, or roughly one thirtieth the length of an adult *C. elegans* (*Figure 1g*, *Videos 1–3*). Moving the camera closer to the WM or using a longer focal length lens reduces the field of view but enables higher resolution images to be attained (*Figure 1h*, *Video 4*).

In humans and diverse model organisms, locomotor activity has been used as a measure of health (*Hausdorff et al., 1997*; *Grotewiel et al., 2005*; *Huang et al., 2004*). It has been shown that spontaneous locomotion on food is a non-ideal measure of health since it assesses food preference in addition to locomotor ability (*Hahm et al., 2015*). As such, a directed behavior is preferable over spontaneous movement. We used the response to blue light illumination as a measure of locomotor ability and therefore health.

The WM can be imaged continuously or intermittently. Under intermittent imaging, the WM is automatically imaged at 0.2 frames per second for a 30 min interval twice daily. After fifteen minutes, a blue light stimulus, which evokes an escape response in worms (*Edwards et al., 2008*), is applied to the entire plate for 10 s using a set of light emitting diodes (LEDs). In this manner we measure both spontaneous and evoked behavioral responses (*Figures 1f* and *2b–e*, *Videos 1–3*).

Custom MATLAB software (MathWorks, Natick, MA) quantifies movement of the animal in each well. Following pixel-by-pixel subtraction of pairs of temporally adjacent frames (*Raizen et al., 2008*), a simple measure of behavior can be defined as the number of pixels whose intensity changes between subsequent image frames (*Figure 2a*). Highly mobile animals yield high activity values, whereas slowly moving, older, or quiescent animals yield low activity values (*Figure 2b–e*). Activity analysis generated rich behavioral data capable of characterizing aging and development (*Figure 2f*) (*Nelson et al., 2013*, *Nelson et al., 2014*; *Iannacone et al., 2017*). Lifespan is determined as the final time of nonzero movement (*Figure 2f*).

During each imaging epoch we calculate the maximum activity before and after the blue light stimulus, and we term these values the spontaneous and stimulated locomotion, respectively. We recorded the total time spent moving after the stimulus, which we term response duration. Finally, we record the response latency, defined as the delay between the end of the stimulus and the beginning of the animal's movement.

Our WorMotel method is designed to be scalable to large numbers of animals. Using standard automation tools including a plate carousel and plate handling robot (*Figure 1—figure supplement*

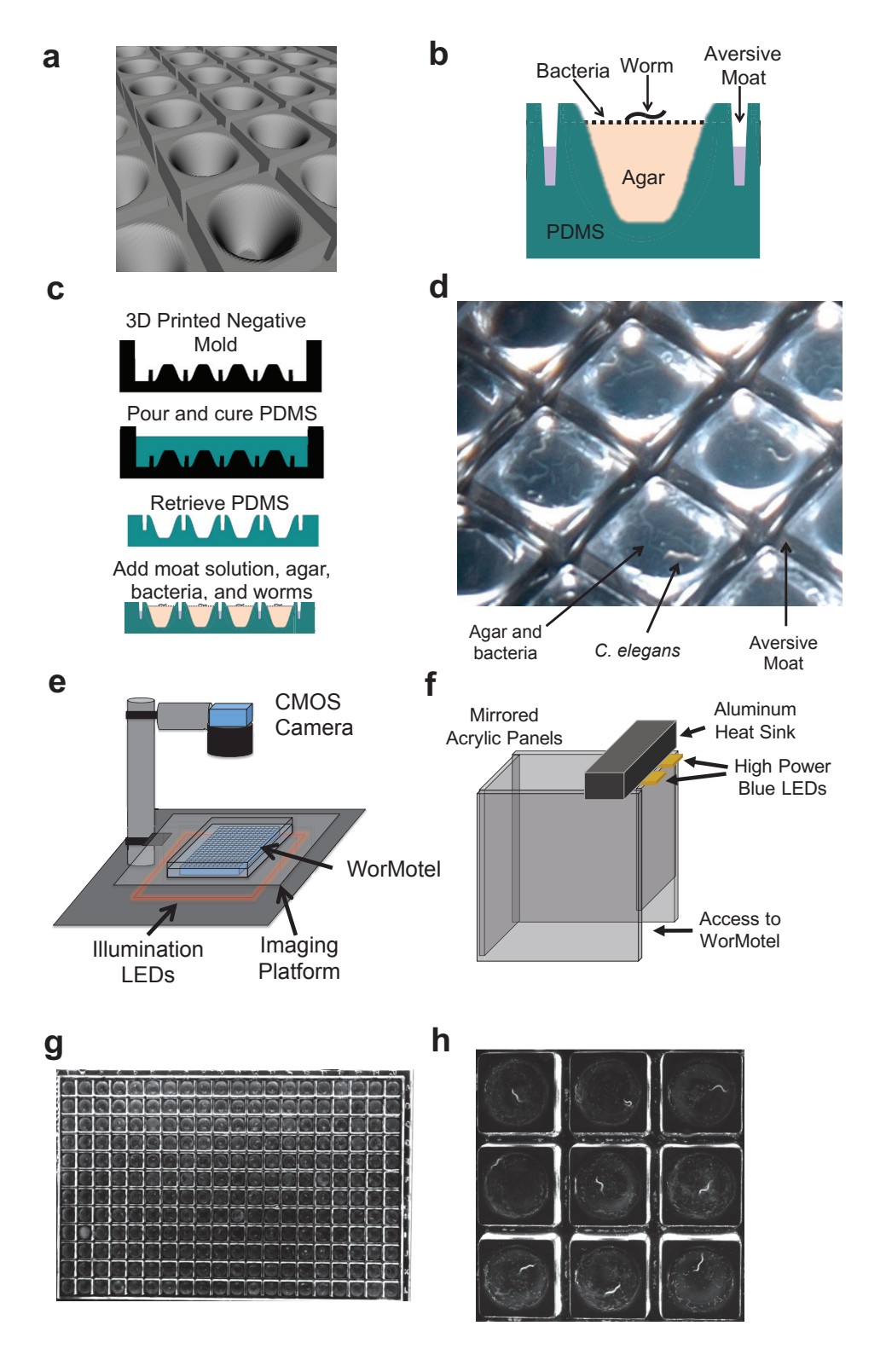

**Figure 1.** WorMotel design, fabrication, preparation, and experimental setup. (a) 3D rendering of WorMotel geometry. (b) Schematic cross-section of a single well. (c) Fabrication and loading process (d) Image of WorMotel filled with agar, bacteria, and adult *C. elegans*. (e) Experimental setup (f) Schematic of blue light stimulation system. (g) Representative image of 240-well WorMotel (h) High resolution image of nine wells, each housing a single young adult N2 worm.
*Figure 1 continued on next page*

*Figure 1 continued*

The following figure supplements are available for figure 1:

**Figure supplement 1.** Comparison of image quality with standard 384-well plate and WorMotel.

**Figure supplement 2.** 240-Well WorMotel (a) 240-well WorMotel PDMS insert.

**Figure supplement 3.** WorMotel prevents burrowing.

**Figure supplement 4.** Schematic of automated imaging system.

*4*) we have developed a system capable of intermittently imaging up to 240 plates containing 57,600 individually tracked worms.

## The WorMotel enables automated lifespan and behavior measurements

We asked to what extent lifespan results from the WM agreed with those using standard manual methods. We compared the survival curves of animals reared on standard agar plates to those in the WorMotel. We measured the lifespans of wild-type N2 alongside the short-lived strain *daf-16(mu86)* (*Ogg et al., 1997*) and the long-lived strain *daf-2(e1370)* (*Kenyon et al., 1993*) at 25° C (*Figure 2g*). For worms grown on standard plates, lifespan assays were carried out using standard methods (*Kenyon et al., 1993*). As expected, mean lifespan of *daf-16* mutants (WM: 7.7 ± 0.3 days, n = 61; Manual: 7.0 ± 0.2 days, n = 117) was shorter than that of N2 (WM: 12.3 ± 0.3 days, n = 123; Manual: 12.2 ± 0.5 days, n = 94) while *daf-2* mutants showed a longer lifespan (WM: 33.5 ± 1.9 days, n = 46, Manual: 30.8 ± 2.0 days, n = 52). We found no significant difference between survival curves acquired from worms grown on standard plates and those grown on the WorMotel. Moreover, our lifespan results agree with those previously reported for each strain (*Kenyon et al., 1993*).

We asked whether the aversive copper sulfate solution in the moat, which helps to retain animals inside their respective wells, had any effect on survival or development. We compared the duration of the L4 stage, a measurement of developmental rate, and lifespan of worms grown on a WorMotel with moats filled with NGMB to a WorMotel with moats filled with 100 mM copper sulfate (see Materials and methods). We found no difference in developmental rate of worms grown from the L3 stage to adulthood in the presence of an NGMB (L4 Duration = 12.2 ± 0.2 hr) and copper sulfate moat (L4 Duration = 12.2 ± 0.2 hr, p=0.95). Similarly, we found no difference in the mean lifespan of N2 worms grown in the presence of an NGMB (19.2 ± 0.7 days, n = 22) or copper sulfate moat (18.9 ± 1.0 days, n = 23).

To test the accuracy of our automated assessment of lifespan, we manually scored the lifespan of worms grown on a WorMotel that was also imaged by camera. Manual assessments of lifespan were performed daily within two hours of a thirty-minute imaging epoch. We compared the time of death measured by our software to that measured by a human observer scoring death manually by standard methods (*Kenyon et al., 1993*). For N2 worms, the average difference between manual and automated lifespan measurement was 0.66 ± 0.6 days (n = 79), indicating that automated lifespan calculation is accurate within the time resolution of standard lifespan assays. Automated lifespan was always less than or equal to manual lifespan, indicating that dead animals were never incorrectly scored as alive. Furthermore, we found no correlation between the error in automatic lifespan measurement and

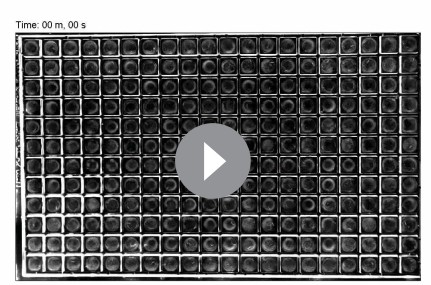

**Video 1.** Young (day 1) adults in the 240-well WorMotel.     A 10 s long blue light exposure (bright flash) occurs at the 15 min mark. FOV: 95 mm x 63 mm

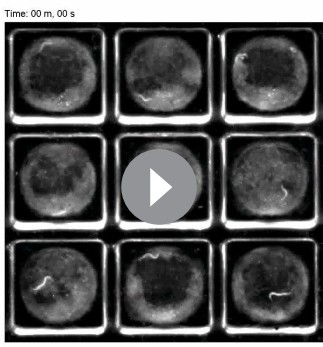

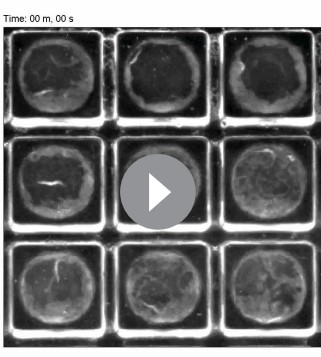

**Video 2.** Detail of 9 wells containing day 1 N2 adults.    A 10 s long blue light exposure (bright flash) occurs at the 15 min mark. FOV: 14 mm x 14 mm

**Video 3.** Detail of 9 wells containing day 10 N2 adults.    A 10 s long blue light exposure (bright flash) occurs at the 15 min mark. FOV: 14 mm x 14 mm

true lifespan (p=0.18), indicating that the error in automated lifespan score is independent of lifespan itself, i.e. absolute measurement error does not increase for worms with longer lifespan.

Together, these results show that with regard to development and lifespan, results from the WorMotel are similar to those using standard methods.

## Mutant strains display diverse behavioral profiles during aging

Aging in *C. elegans* is accompanied by a deterioration of many behaviors, including slowing of locomotion (*Hahm et al., 2015*) and feeding (*Huang et al., 2004*), and a reduced capacity for learning and memory (*Stein and Murphy, 2012*). While many genes have been identified that regulate aging, less is known about the effect of these genes on behavior.

To survey the relationship between lifespan and behavior, we used the WM to analyze behavioral trajectories for wild type worms and seven loss-of-function mutants for genes known to influence lifespan and/or behavior: (1) *daf-2,* which encodes the insulin/IGF receptor (*Kenyon et al., 1993*), (2) *age-1*, which encodes phosphoinositide-3-kinase, a component of the insulin/insulin-like signaling (IIS) pathway (*Friedman and Johnson, 1988*), (3) *daf-16*, which encodes a transcription factor regulating a cytoprotective response (*Ogg et al., 1997*), (4) *tax-4*, which encodes a cyclic nucleotide-gated channel required for some sensory transduction (*Apfeld and Kenyon, 1999*), (5) *unc-31*, required for neuropeptide processing (*Ailion et al., 1999*), (6) *lite-1*, a gene encoding a receptor required for normal aversive response to blue light (*Edwards et al., 2008*), and (7) *aak-2*, a gene encoding a subunit of AMP kinase (*Apfeld et al., 2004*).

We assayed the activity and survival of individuals within populations of each strain (*Figure 3a–f*, *Figure 3—figure supplement 1*). Mean lifespans of mutant strains relative to wild type agreed with those reported in previous studies (*Table 1*). The shapes of behavior curves varied dramatically between strains.

We included *lite-1* worms in order to determine whether these mutants, previously shown to have a reduced response to blue light (*Edwards et al., 2008*; *Ward et al., 2008*), can be assayed by our blue light illumination system

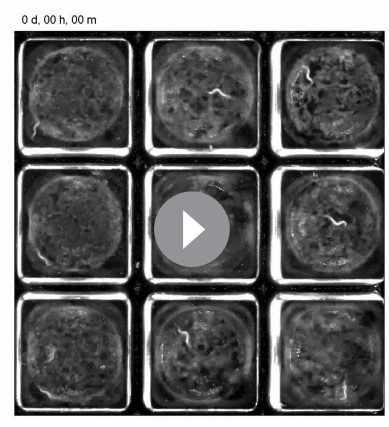

**Video 4.** High resolution video of 9 WorMotel wells containing adult animals (day 1–4).    FOV: 14 mm x 14 mm

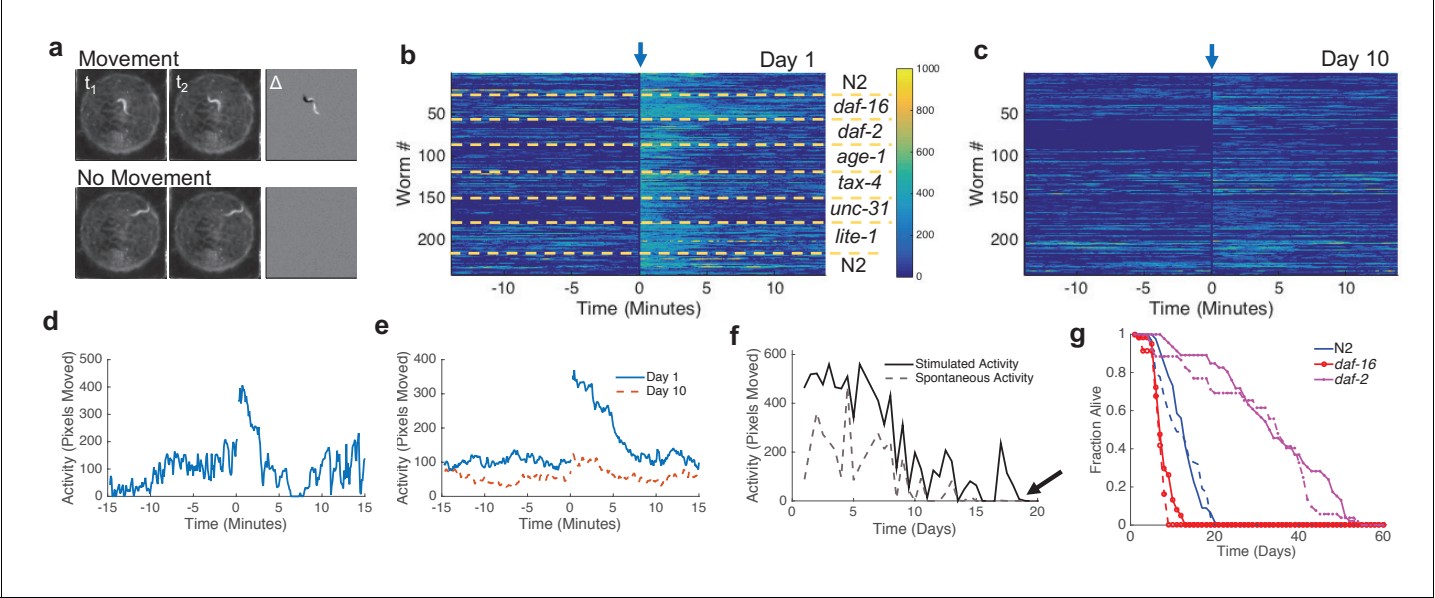

**Figure 2.** Image processing, automated lifespan calculation, and manual validation. (a) Activity calculation. Delta image (right image) is calculated by pixel-by-pixel subtraction of images taken at time $t_1$ from images taken at time $t_2$. Examples are shown in which a worm moves (top) and does not move (bottom). (b) Activity, shown as a heat map, of young adult (day one) animals during a 30-min imaging epoch. Arrow indicates 10-s blue light stimulation. Color bar indicates the number of pixels changed over a 60-s interval. (c) Activity of adult *C. elegans* on day 10. Colorbar uses same scale as panel (b). (d) Activity trace from one wild-type animal. Blue light was applied at time zero. (e) Average wild-type population behavior on day 1 (solid curve) and day 10 (dashed curve) (n = 30). (f) Single animal trace of maximum spontaneous (dashed) or stimulated (solid) activity across entire lifespan. Arrow indicates time of death. (g) Lifespan of animals grown on the WorMotel (solid curves) or standard plates (dashed curves).

for measuring stimulated activity. To our surprise, we found little difference in the responses to blue light between N2 and *lite-1* mutants, possibly because our light stimulus is higher in irradiance than that previously used, and/or our activity measurement is more sensitive to movement increases than the previously measured body bend frequency (*Edwards et al., 2008*). Regardless, our results for *lite-1* worms indicate that even mutants with deficits in blue light response can be assessed by our method.

Together, these results show that the WorMotel method is compatible with arbitrary strains. We used the longitudinal data generated by the WorMotel to address questions about the relationship between aging and behavior.

### Mutants with short and long lifespan display patterns of late-life behavioral decline resembling those of short and long-lived worms from a wild-type population

While many factors are known to modulate the mean lifespan of a population, less is known about how these factors alter the aging process on an individual level. Zhang *et al.* recently showed that within a wild-type population, long-lived and short-lived animals differed in two ways (*Zhang et al., 2016*). First, the rate of physiological decline was slower in long-lived individuals, as might be expected. The second, however, was counter-intuitive: the additional lifespan of longer-lived individuals was primarily due to differences toward the end of the lifespan. That is, long-lived animals exhibited longer periods of low physiological function, or 'extended twilight' (*Zhang et al., 2016*).

A different picture was suggested by a study using automated assays of lifespan in the 'Lifespan machine' (*Stroustrup et al., 2016*). In this study it was reported that various genetic and environmental perturbations do not fundamentally change the shape of the survival curve, but rather only compress or dilate it in time. This result was interpreted as suggesting that the aging process in *C. elegans* is, at least at some point in its pathway, controlled by a single process describable by a single variable corresponding to the rate of aging (*Stroustrup et al., 2016*).

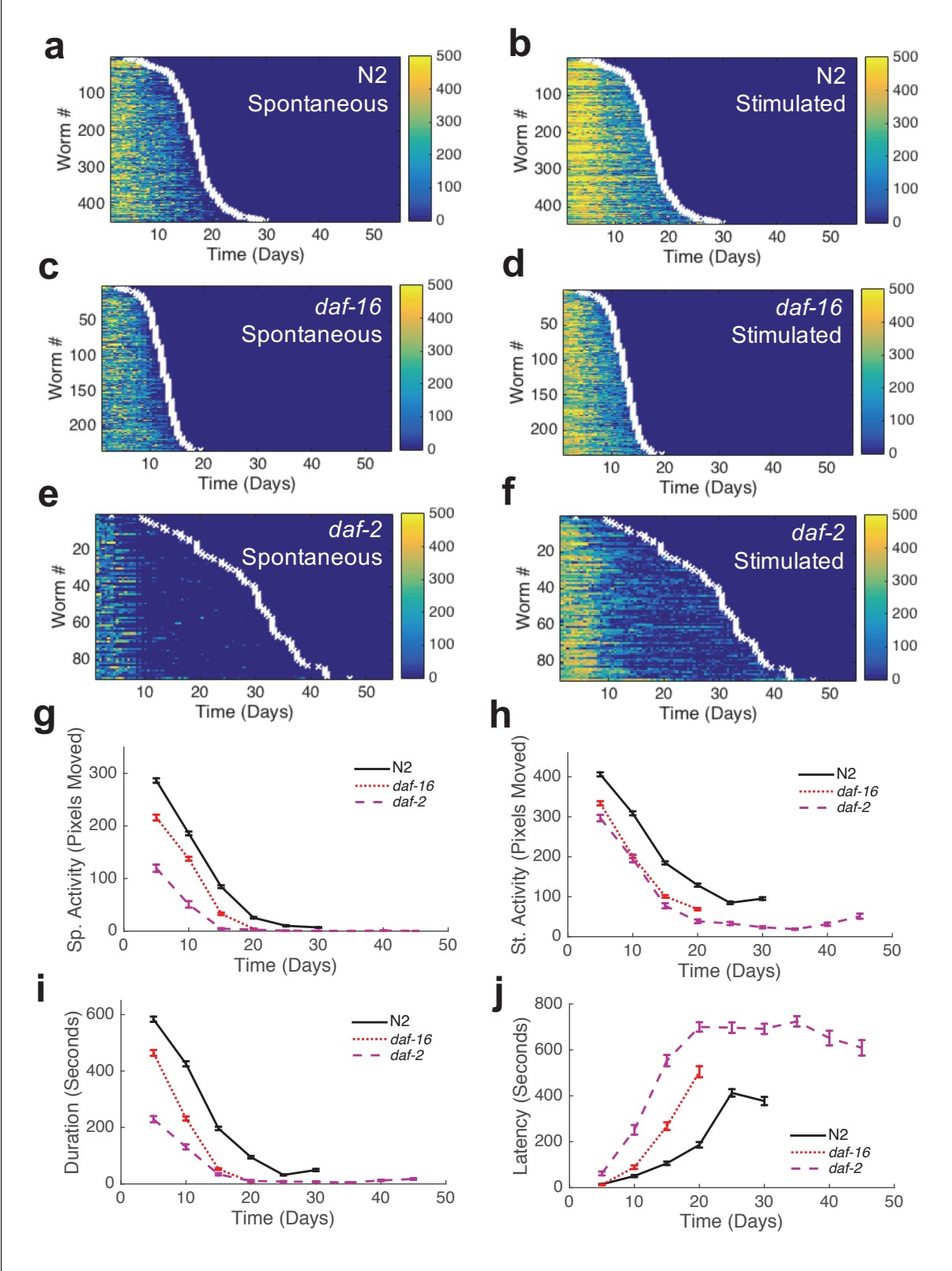

**Figure 3.** Automated quantification of behavioral changes during aging. (**a**) Spontaneous behavior heat map for N2 (n = 445). Color bar indicates the number of pixels changed over a 60-s interval. (**b**) Stimulated behavior heat map for N2. (**c**) Spontaneous behavior heat map for *daf-16* (n = 234). (**d**) Stimulated behavior heat map for *daf-16*. (**e**) Spontaneous behavior heat map for *daf-2* (n = 90). (**f**) Stimulated behavior heat map for *daf-2*. (**g**) Survivor

*Figure 3 continued on next page*

*Figure 3 continued*
population spontaneous activity. (h) Survivor population stimulated activity. (i) Survivor population response duration. (j) Survivor population response latency.
The following source data and figure supplement are available for figure 3:

**Source data 1.** Includes data for lifespan, spontaneous locomotion, stimulated locomotion, response duration, and response latency for each strain shown.
**Figure supplement 1.** Additional behavior heat maps.

We sought to determine to what extent 'extended twilight' and/or scaling effects apply at the behavioral level in mutants with altered aging. The concept of a universal scaling parameter in aging would suggest that the short and long-lived individuals within any strain (whether with normal, short, or long mean lifespan) would resemble their short and long-lived counterparts in the reference strain, but with a temporal scaling (*Figure 4a*). If the variations in aging rate among individuals in any isogenic strain are governed by similar factors, we would expect that short and long-lived individuals would display similar late-life characteristics as their wild type counterparts. If, on the other hand, short-lived strains as a whole physiologically more closely resemble short-lived individuals of a wild type population, we might expect them to display late-life characteristics similar to these short-lived individuals (*Figure 4b*). Similarly, long-lived strains might display a range of late-life decays or alternatively collectively resemble long-lived worms in the reference strain.

Wild-type strain N2 worms exhibited an initial decline followed by a 'plateau' period of nearly constant spontaneous and stimulated activity and response duration and latency (*Figure 3g–j*). When we compared the behavior of the shortest-lived and longest-lived quartile of N2 worms, we found that their behavioral declines were qualitatively different. The longest-lived animals exhibited a 'decline and plateau' phenotype, in which an initial rapid decline in behavioral capacity is later replaced by a very gradual decline for the remainder of life (*Figure 5a,h*). By contrast, the shortest-lived animals showed only the rapid decline in behavior before dying (*Figure 5a,g*). The result that

**Table 1.** Summary of lifespan data

| Strain | Food source | Lifespan (Mean ± SD) (Days) | N | P-value relative to N2 control |
|---|---|---|---|---|
| N2 | DA837 | 16.8 ± 4.4 | 445 | N/A |
| *daf-16* | DA837 | 12.4 ± 2.6 | 234 | $3.0 \times 10^{-45}$ |
| *daf-2* | DA837 | 28.5 ± 9.2 | 90 | $2.1 \times 10^{-16}$ |
| *age-1* | DA837 | 23.7 ± 8.9 | 119 | $4.3 \times 10^{-9}$ |
| *tax-4* | DA837 | 22.0 ± 5.4 | 72 | $3.6 \times 10^{-6}$ |
| *unc-31* | DA837 | 21.2 ± 7.2 | 120 | $3.6 \times 10^{-4}$ |
| *lite-1* | DA837 | 19.7 ± 4.0 | 90 | 0.036 |
| *aak-2* | DA837 | 9.6 ± 1.8 | 117 | $3.9 \times 10^{-30}$ |
| N2 | HT115 (EV RNAi) | 20.9 ± 10.5 | 80 | N/A |
| N2 | HT115 (*daf-2* RNAi) | 29.9 ± 10.3 | 38 | $1.2 \times 10^{-6}$ |
| N2 | HT115 (*odr-10* RNAi) | 21.5 ± 6.5 | 40 | 0.53 |
| *daf-2* | HT115 (EV RNAi) | 30.0 ± 10.5 | 40 | $1.6 \times 10^{-6}$ |
| *daf-2* | HT115 (*daf-2* RNAi) | 34.5 ± 9.5 | 39 | $7.9 \times 10^{-11}$ |
| *daf-2* | HT115 (*odr-10* RNAi) | 36.9 ± 7.4 | 39 | $8.2 \times 10^{-15}$ |
| *odr-10* | HT115 (EV RNAi) | 21.5 ± 6.6 | 40 | 0.40 |
| *odr-10* | HT115 (*daf-2* RNAi) | 28.3 ± 12.5 | 38 | $4.8 \times 10^{-4}$ |
| *odr-10* | HT115 (*odr-10* RNAi) | 21.1 ± 7.5 | 40 | 0.49 |

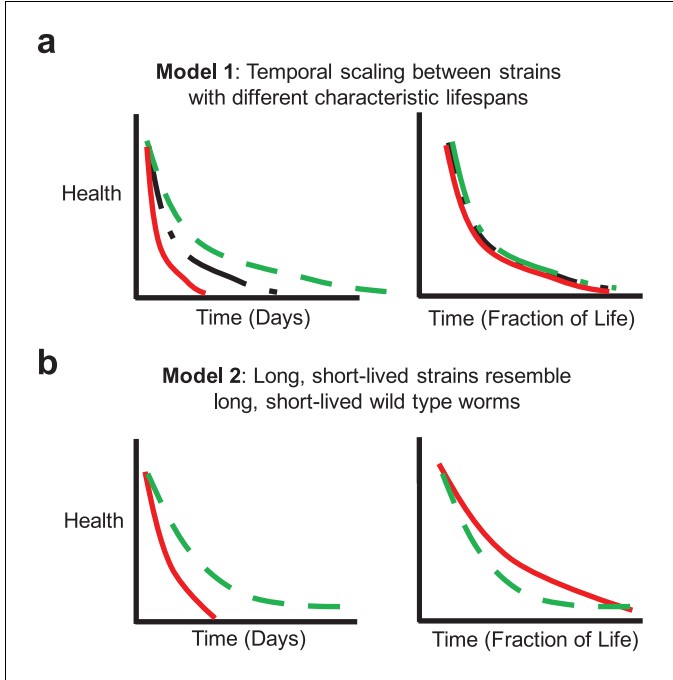

**Figure 4.** Potential aging models. (**a**) Model 1: Temporal scaling results in identical patterns of behavioral decline when data are normalized by lifespan. Idealized decline curves for wild type (black dot-dashed), short-lived (red solid), and long-lived (green dashed) strains. Decline curves are shown as a function of chronological time (left) and fraction of life (right). (**b**) Model 2: Long-lived and short-lived strains resemble long-lived and short-lived wild type worms with respect to behavioral decline. Idealized decline curves for a short-lived (red solid) and a long-lived (green dashed) strain. Decline curves are shown as a function of chronological time (left) and fraction of life (right).

long-lived animals experience a long period of low behavior are consistent with the 'extended twilight' reported by Zhang et al. (*Zhang et al., 2016*).

Short-lived *daf-16* mutants declined at a similar rate to N2, but did not exhibit any plateau phase; instead, *daf-16* worms die after their initial behavioral decline (*Figure 3g–j*, *Figure 5c,d*). A similar effect was seen in *daf-16* response duration and response latency, which do not level off but decrease or increase, respectively, at a similar rate until the time of death. Comparing the activity history of the shortest-lived N2 worms to that of *daf-16* as a whole, we found a striking correspondence between the behavioral decline of the two groups (*Figure 5g*). These results show that the behavioral decline of *daf-16* animals is not a scaled version of the wild type distribution of decline, but instead resembles the short-lived individuals in a wild-type population.

Long-lived *daf-2(e1370)* mutants, in which behavioral quiescence has been previously reported (*Gems et al., 1998*; *Gaglia and Kenyon, 2009*), exhibited a decline in stimulated activity akin to that observed in N2 and *daf-16* followed by a nearly constant low level of stimulated activity and response behaviors for the remainder of life (*Figure 3h*). Spontaneous activity in *daf-2*, on the other hand, declined to near zero within 10 days of adulthood, where it remained until death. Even at very young chronological age (before day 5), *daf-2* mutants perform less well than N2 for each behavior metric scored (*Figure 3g–j*).

The 'decline and plateau' phenotype of the longest-lived N2 animals was also evident in both short-lived and long-lived *daf-2* animals (*Figure 5e,f*). Long-lived strains *age-1*, *tax-4*, and *unc-31* also exhibited the 'decline and plateau' phenotype (*Figure 3—figure supplement 1*, *Figure 5—figure supplement 1*). These results show that aging behavior of *daf-2* and other long-lived animals, like that of *daf-16* animals, does not resemble a scaled version of wild type. Instead, they resemble the longest-lived individuals in a wild-type population, in that they exhibit a long plateau period of low locomotory function during late life.

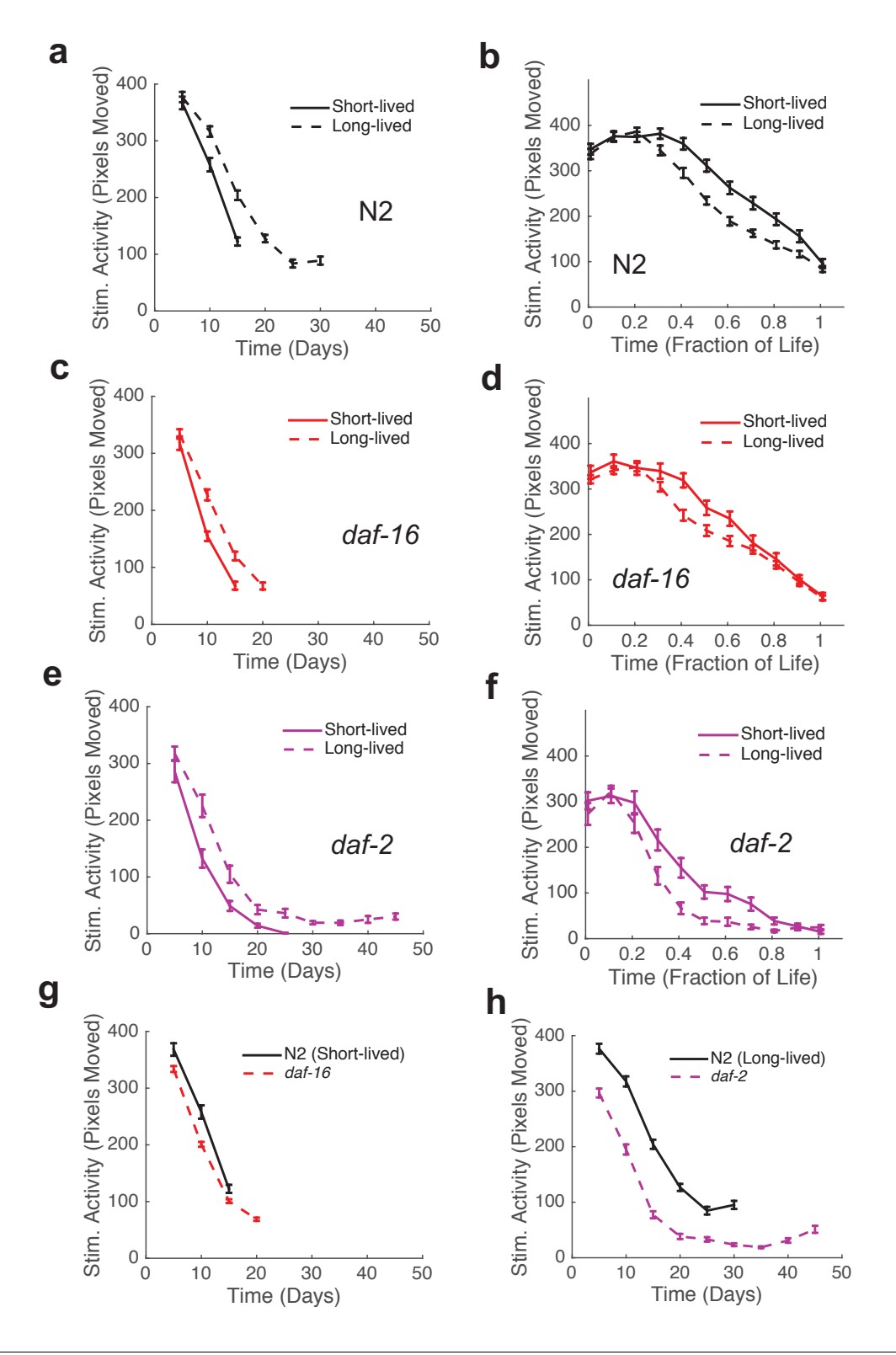

**Figure 5.** Mutants with short and long lifespan display patterns of late-life behavioral decline that resemble short and long-lived worms from a wild type population. (a) N2 behavior over time for the lowest quartile (solid curve) and highest quartile (dashed curve) of survivors. (b) Data from panel (a) plotted as a fraction of each individual's life. (c) *daf-16* behavior over time. (d) *daf-16* behavior over fraction of life. (e) *daf-2* behavior over time. (f) *daf-2* behavior over fraction of life. (g) Comparison of N2 lowest survivor quartile and *daf-16*. (h) Comparison of N2 highest survivor quartile with *daf-2*.
*Figure 5 continued on next page*

*Figure 5 continued*

The following figure supplement is available for figure 5:

**Figure supplement 1.** Stimulated activity versus chronological time and fraction of life for long-lived individuals and short-Lived individuals of the same strain.

In order to further characterize inter-individual differences in aging, we next sought to quantify the shape of behavioral decline. We analyzed individual behavioral decline as a fraction of life and calculated early and late decline rates (*Figure 5b,d,f*, *Figure 5—figure supplement 1*, *Figure 6a–c*) (see Materials and methods). We then calculated the difference in decline rates to quantify the overall shape of behavioral decline. We found that the change in decline rate negatively correlated with lifespan in all strains tested (N2: R = −0.32, p=8.0×10⁻¹²; *daf-16*: R = −0.18, p=0.0054; *daf-2*: R = −0.25, p=0.016; *age-1*: R = −0.43, p=8.0×10⁻⁷; *tax-4*: R = −0.41, p=4.2×10⁻⁴; *unc-31*: R = −0.49, p=1.6×10⁻⁸, *lite-1*: R = −0.54, p=3.9×10⁻⁸, *aak-2*: R = −0.23, p=0.012) (*Figure 6a–e*), indicating that the shape of behavioral decline differed signficantly for individuals with differing lifespans. We observed a smooth transition between the shape of aging behavior between short-lived and long-lived individuals within each strain (*Figure 6a–c*, *Figure 6—figure supplement 1*). Furthermore, our results suggest that there exists a relationship between change in decline rate and lifespan that lies along a continuum across strains in addition to between individuals of the same strain (*Figure 6e,f*). Therefore, our results suggest that while behavioral decline does not temporally scale with lifespan, the stochastic sources of variability between isogenic individuals modulate the shape of aging along the same axis of variability as between short and long lived strains. For example, variability in the rate of aging may reflect a variability in the nuclear localization of DAF-16 and the activation of its targets.

Finally, we investigated the level of inter-individual variability in the rate of aging. We found that the standard deviation of decline rate change generally decreased with average lifespan (*Figure 6g*) (R = −.70, p=0.055). That is, longer-lived strains exhibited less individual variability than shorter-lived strains. Under a temporal scaling model, both the mean decline rate change and standard deviation of decline rate change would be equal across strains with different lifespans. Therefore, our data argue against a temporal scaling model of aging.

## Behavioral decline during acute oxidative stress resembles behavioral decline during aging

In an analysis of scaling in lifespan, Stroustrup *et al.* showed that the shape of population survival curves during thermal stress was virtually identical to that occurring during aging (*Stroustrup et al., 2016*). That is, the rate of population decline increases with temperature while still obeying the same fundamental kinetics. We have shown that the shape of behavioral decline is not identical for worms with different lifespans within a population. Nevertheless, we asked whether a similar scaling law holds for behavioral decline in populations during acute stress.

To test this idea, we added paraquat, which induces oxidative stress via generation of reactive oxygen species (ROS), to the WorMotel agar and monitored the animals' subsequent behavior and survival (*An et al., 2003*). Oxidative stress, like thermal stress, greatly shortens lifespan. We found that when we added paraquat to a final concentration of 40 mM on day 1 of adulthood, wild type animals survived for 21.2 ± 8.9 hr (n = 58), consistent with previous results (*Figure 7a*, *Table 2*) (*An et al., 2003*). We compared the shape of decline for animals experiencing stress to animals experiencing normal aging (*Figure 7b,d,f,h*). We calculated the normalized mean square difference between behavior during aging and stress, and, after correcting for the average activity offset, found that the difference was 4.1% for N2, 9.7% for *daf-16*, 11.0% for *daf-2*, and 13.2% for *age-1*.

If it is true that behavioral decline during stress beginning at day 1 of adulthood resembles a temporally scaled recapitulation of behavioral decline during normal aging, we reasoned that starting oxidative stress at mid-life should truncate the initial portion of the behavioral decline curve, as the slow decline of early aging should have already occurred naturally. To test this idea, we grew animals on the WorMotel as we did for normal aging experiments. We then added paraquat on day 9 of

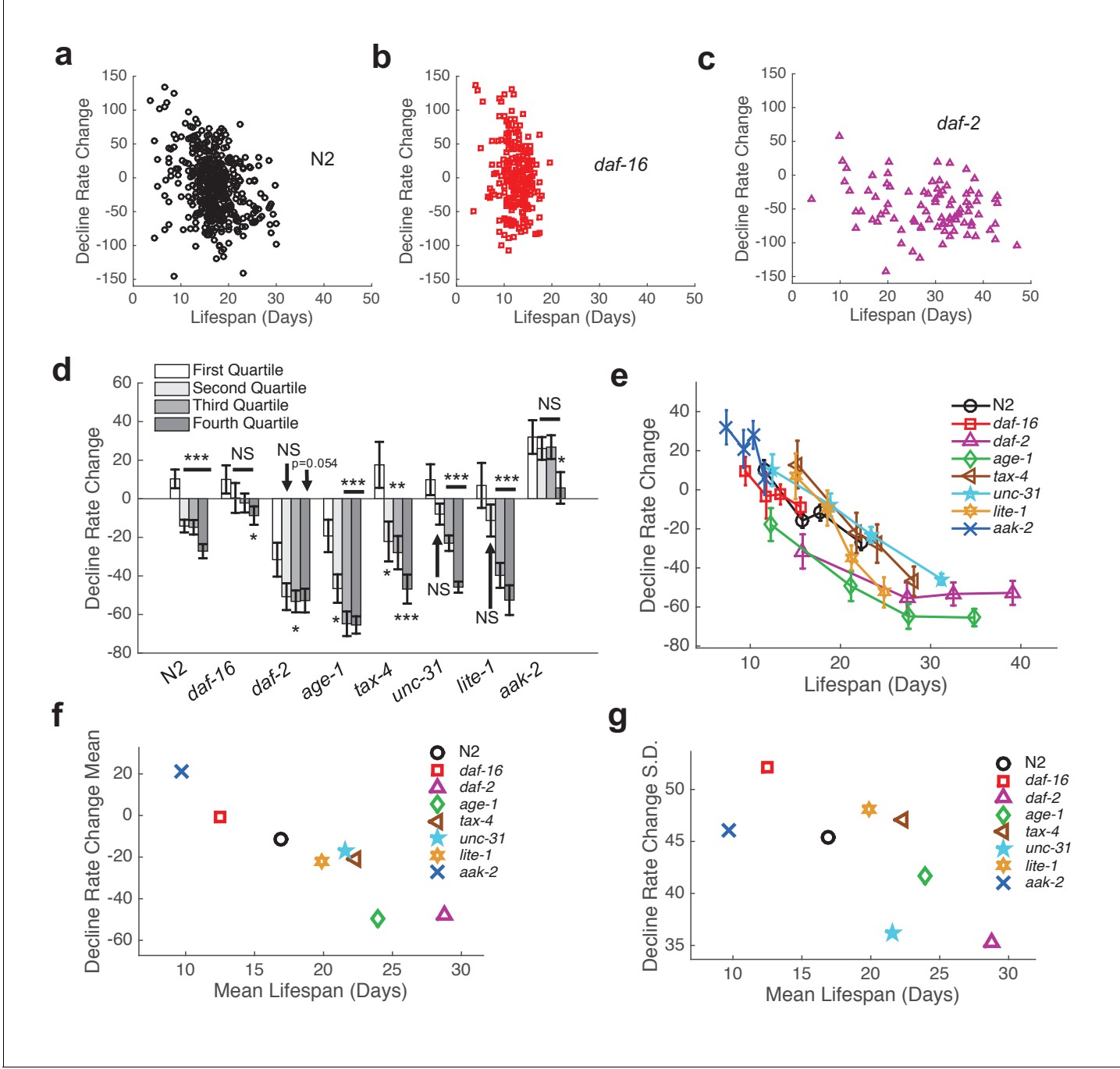

**Figure 6.** Shape of behavioral decline changes continuously with lifespan across individuals and strains. (a) Change in decline rate (pixels/life fraction) versus lifespan for individual N2 animals. (b) Same data as in panel (a) presented for *daf-16* mutants. (c) Same data as in panel (a) presented for *daf-2* mutants. (d) Change in decline rate (pixels/life fraction) in lowest (white) to highest (dark gray) survivor quartiles. *, p<0.05; **, p<0.01; ***, p<0.001. (e) Change in decline rate (pixels/life fraction) versus lifespan for multiple strains. (f) Mean decline rate change (pixels/life fraction) plotted against mean lifespan for each strain tested. Correlation coefficient r = −0.94, p=0.0006. (g) Standard deviation of decline rate change (pixels/life fraction) plotted against mean lifespan for each strain tested. Correlation coefficient r = −0.70, p=0.055.

The following figure supplement is available for figure 6:

**Figure supplement 1.** Change in decline rate versus lifespan in individual animals.

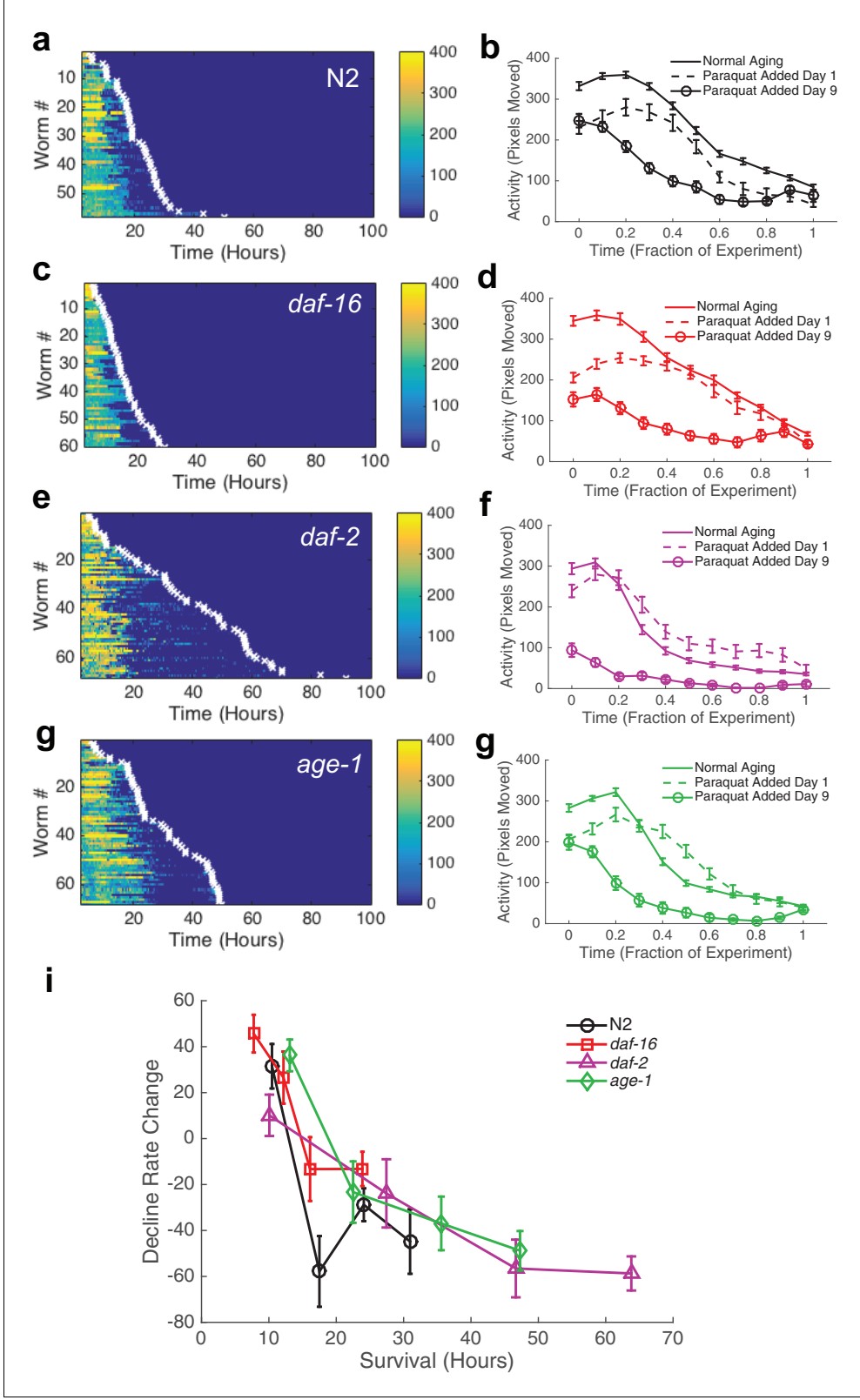

**Figure 7.** Behavioral decline during acute oxidative stress resembles behavioral decline during aging. (a, c, e, g) Behavior and survival heat maps for N2 (n = 58), *daf-16* (n = 60), *daf-2* (n = 68), and *age-1* (n = 68). (b, d, f, h) Comparison of behavioral decline during normal aging (solid curve), stress with paraquat added on day 1 of adulthood (dashed curve), and stress with paraquat added on day 9 of adulthood (solid curve with circles) (i) Change in decline rate (pixels/life fraction) for individuals versus survival on paraquat.

*Figure 7 continued on next page*

*Figure 7 continued*

The following source data is available for figure 7:

**Source data 1.** Includes activity and paraquat survival data for each strain shown.

adulthood, and monitored worms' subsequent behavior and survival. We observed that the initial decline occurred almost immediately, indicating that, as expected, the initial portion of the aging curve was no longer present in the behavioral stress decline curve (*Figure 7b,d,f,h*). These results suggest that population-level rate of behavioral decline indeed temporally scales with increased stress.

Previously, we observed that during normal aging, the change in decline rate, a measure of the shape of functional decline, negatively correlated with lifespan. Since population behavioral decline seemed to scale between normal aging and stress, we therefore investigated the relationship between individual aging and survival during stress. We once again found a negative correlation between the change in decline rate and survival for animals grown on 40 mM paraquat (*Figure 7i*).

These results show that the shape of behavioral decline during severe oxidative stress was similar to that during and aging, despite the process of aging on oxidative stress occurring about 20 times faster. This result suggests that there exist strong parallels in the worm's behavioral responses to oxidative stress and to aging.

## ODR-10 is required for elevated response threshold but not increased lifespan of *daf-2* mutants

We observed that long-lived *daf-2* mutants exhibited greatly reduced locomotion amplitude and movement duration and elevated response latency to aversive blue light (*Figure 3g–j*). Previous studies have observed similar phenotypes, such as the high degree of dauer-like quiescence in *daf-2* adults (*Gems et al., 1998*; *Gaglia and Kenyon, 2009*). In addition to having increased longevity, *daf-2* mutants have been shown to possess greater fat stores (*Ogg et al., 1997*). Reduction of insulin signaling, higher fat stores, and reduced movement are all features of hibernation in mammals, and it has been proposed that the *daf-2* mutation confers a constitutive 'hibernation-like' phenotype on these animals (*Carey et al., 2003*; *Gaglia and Kenyon, 2009*). Since animals deprived of fat stores or forced to move during hibernation have reduced survival (*Reeder et al., 2012*), we hypothesized that reduced locomotor behavior might be required for increased lifespan in *daf-2* animals.

A recent study identified ODR-10, a G-protein coupled olfactory receptor sensitive to diacetyl, as required for reduced locomotion in *daf-2* mutants (*Hahm et al., 2015*). ODR-10 mRNA levels are elevated in *daf-2* mutants, and *daf-2* mutants on *odr-10* RNAi show a greater maximum velocity than controls. In an effort to determine if reduced locomotion was required for increased lifespan, we tested *daf-2* mutants with *odr-10* RNAi with the WorMotel.

We grew N2, *daf-2(e1370)*, and *odr-10(ky32)* mutants on Empty Vector (EV), *daf-2*, or *odr-10* RNAi on the WorMotel to monitor behavior and lifespan. To our surprise, we found no significant difference in spontaneous activity between either N2 or *daf-2* worms grown on EV versus *odr-10* RNAi (*Figure 8a*). These results suggest that ODR-10 does not in fact influence the reduced spontaneous movement observed in *daf-2* mutants.

**Table 2.** Summary of paraquat assay survival data.

| Strain | Food source | Lifespan (Mean ± SD) (Days) | N | P-value relative to N2 control |
|---|---|---|---|---|
| N2 | DA837 + 40 mM paraquat | 21.2 ± 8.9 | 58 | N/A |
| *daf-16* | DA837 + 40 mM paraquat | 16.5 ± 12.0 | 60 | 0.001 |
| *daf-2* | DA837 + 40 mM paraquat | 37.4 ± 22.0 | 68 | $2.0 \times 10^{-5}$ |
| *age-1* | DA837 + 40 mM paraquat | 29.7 ± 13.8 | 68 | $2.7 \times 10^{-5}$ |

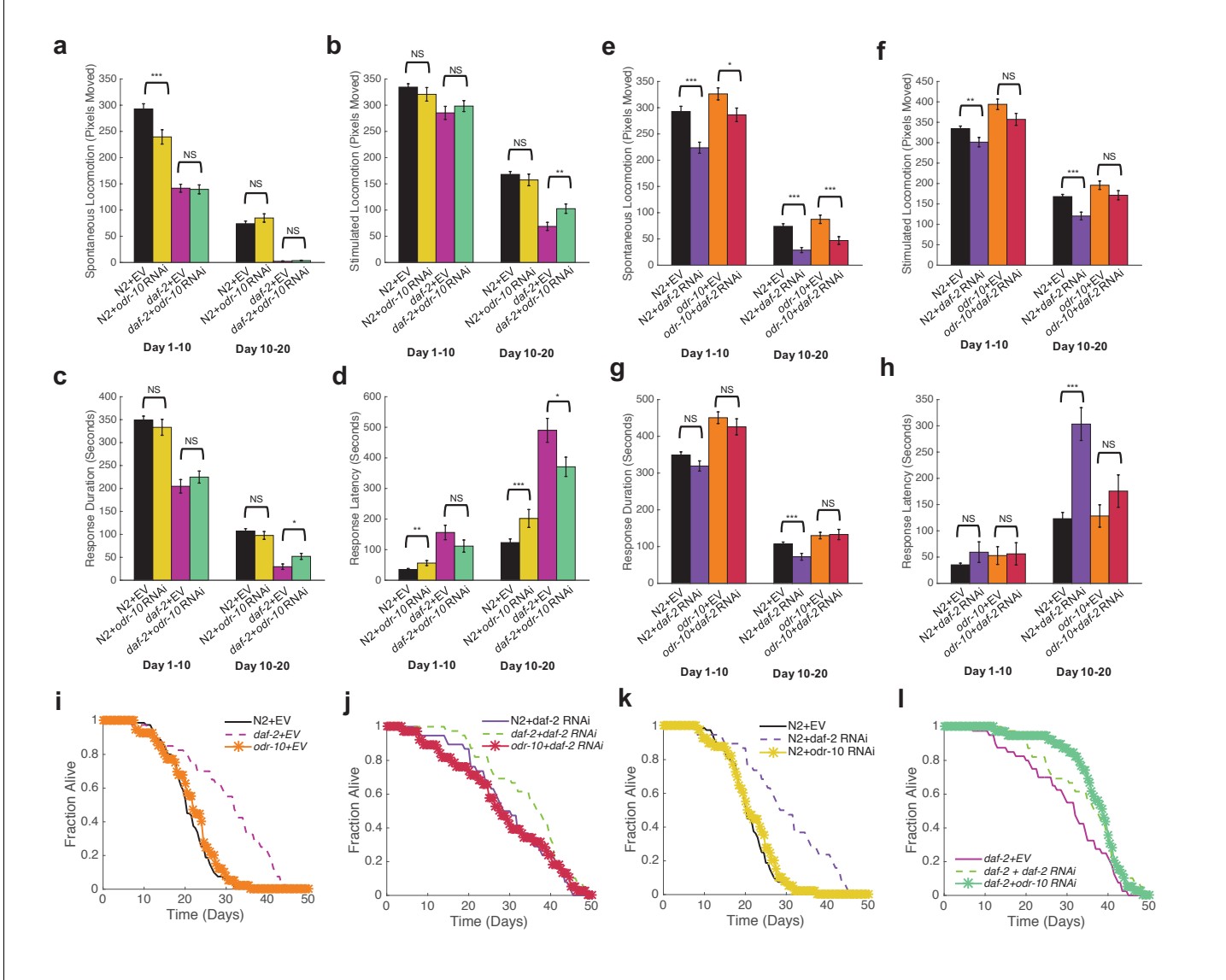

**Figure 8.** Reduced sensory response is not required for extended longevity in *daf-2* mutants. (a) Spontaneous activity during days 1–10 and 10–20 of adulthood for N2 grown on Empty Vector RNAi (n = 80), N2 grown on *odr-10* RNAi (n = 40), *daf-2* grown on Empty Vector RNAi (n = 40), and *daf-2* grown on *odr-10* RNAi (n = 39). *p<0.05; **p<0.01; ***p<0.001. (b) Stimulated activity for the same individuals shown in panel (a). (c) Response duration for the same individuals shown in panel (a). (d) Response latency for the same individuals shown in panel (a). (e) Spontaneous activity during days 1–10 and 10–20 of adulthood for N2 grown on Empty Vector RNAi (n = 80), N2 grown on *daf-2* RNAi (n = 38), *odr-10(ky32)* mutants grown on Empty Vector RNAi (n = 40), and *odr-10(ky32)* mutants grown on *daf-2* RNAi (n = 38). *p<0.05; **p<0.01; ***p<0.001. (f) Stimulated activity for the same individuals shown in panel (e). (g) Response duration for the same individuals shown in panel (e). (h) Response latency for the same individuals shown in panel (e). (i–l) Survival curves.

The following source data is available for figure 8:

**Source data 1.** Includes data for lifespan, spontaneous locomotion, stimulated locomotion, response duration, and response latency for each strain shown.

We examined stimulated activity in the same experiments. While we found no difference in early life stimulated activity between N2 or *daf-2* mutants grown on EV versus *odr-10* RNAi, we did find that late life stimulated activity was elevated in *daf-2* mutants grown on *odr-10* RNAi, whereas late life stimulated activity was unchanged for N2 grown on *odr-10* RNAi (*Figure 8a,b*). Furthermore, while N2 treated with *daf-2* RNAi exhibited significantly decreased stimulated activity compared to

animals grown on EV RNAi, *odr-10* mutants grown on *daf-2* RNAi did not (*Figure 8f*). Similar results were observed for response duration and latency (*Figure 8c,d,g,h*). These results suggest that ODR-10 does not directly affect locomotion in *daf-2* mutants, but reduces sensitivity of *daf-2* mutants to stimuli.

Loss of function mutations in a number of genes required for sensory responses, such as such as the cyclic nucleotide channel TAX-4 and the intraflagellar transport particle homologue OSM-6, have been shown to exhibit extended longevity (*Apfeld and Kenyon, 1999*). Because *daf-2* mutants exhibit both reduced locomotion in addition to prolonged lifespan, it is possible that the reduced sensory responsiveness of *daf-2* mutants is a requirement for their extended longevity.

To test this idea, we compared survival curves for worms in which either *daf-2, odr-10*, or both either had loss of function mutations or were knocked down by RNAi. We found that *odr-10* mutation and *odr-10* RNAi had no effect on wild type lifespan (*Figure 8i,k*), whereas *daf-2* RNAi increased lifespan in both wild type and *odr-10* mutants to an equal degree (*Figure 8j,l*). These results show that while ODR-10 is required for reduced sensory responses in *daf-2* mutants, it is not required for extended lifespan.

## Discussion

### A powerful and flexible tool for assaying *C. elegans* behavior

We have used our WorMotel method to investigate inter-individual and inter-strain variability in behavioral decline and the relationship between behavior and lifespan. Large-scale automated analysis of lifespan and behavior will facilitate screening for genetic and chemical modulators of aging. This system's ability to longitudinally monitor animals throughput their lifespans may also help identify mechanisms of variability of aging between individuals in a population.

In addition to these applications in aging, we have used the WorMotel in studies of other behaviors, such as developmentally-timed quiescence (lethargus) (*Nelson et al., 2013*), stress-induced quiescence (*Iannacone et al., 2017*; *Nelson et al., 2014*), and adult behavior states (*McCloskey et al., 2017*). Therefore the WorMotel is a flexible tool for assaying *C. elegans* long-term behavior.

### Functional decline follows a universal dependence on lifespan in diverse strains

We found that short-lived wild type individuals declined in a manner similar to the short-lived strain *daf-16* and that long-lived wild type individuals declined in a manner similar to the long-lived strain *daf-2*. Furthermore, we found that the shape of decline followed a single curve with respect to lifespan (*Figure 6e*).

These results suggest that the sources of variability in lifespan in individuals also impact functional decline in a corresponding manner. For example, the N2 worm that survives 15 days due to stochastic factors will decline in a similar manner to the *daf-16* worm that survives 15 days. Furthermore, individuals with a 30-day lifespan will exhibit a different shape of functional decline, but this shape is dictated by the confluence of genetic and stochastic factors that result in the lifespan of 30 days.

One explanation for the extended longevity of insulin signaling mutants such as *daf-2* and *age-1* is via their shared transcriptional profile with dauer larvae, which can persist in harsh environments by virtue of an upregulation in stress-response and detoxification pathways (*McElwee et al., 2004*). Furthermore, other work has shown that at an advanced age, wild type transcriptional profiles also exhibit similarities with that of dauer larvae (*Lund et al., 2002*). Our finding that behavioral decline adheres to a continuum suggests that long-lived wild type worms may be physiologically and transcriptionally similar to worms with mutations in the insulin signaling pathway. Future experiments comparing gene expression in rapidly or slowly aging worms may elucidate how aging variability is manifest at the molecular level.

### Extension of late-life decrepitude in long-lived worms

In addition to this and other studies (*Zhang et al., 2016*), a recent study (*Podshivalova et al., 2017*) also observed an extension of late-life behavioral quiescence in N2 and *daf-2* mutants. The authors found that intestinal bacterial colonization is a risk factor for death in *C. elegans* and that *daf-2* mutants, which exhibit a greater fraction of late-life decrepitude compared to N2, were less

susceptible to this bacterial colonization. When the authors fed worms killed bacteria, they found a greater lifespan extension in N2 (40%) than *daf-2(e1368)* (16%), suggesting that bacterial colonization is a cause of premature death in N2 worms. Finally, the authors found that feeding worms dead bacteria specifically extended the period of infirmity rather than that of good health, suggesting that bacterial colonization may be a primary cause of lifespan truncation in short-lived individuals. That is, bacterial colonization may cause a reduction in late-life decrepitude in short-lived worms by causing premature death.

If bacterial colonization is a fundamental cause of the smaller fraction of late-life decrepitude observed in short-lived worms, the question remains as to why long-lived worms exhibit extended behavioral quiescence in old age. One possibility is that muscle integrity degrades with age faster than other tissues (*Herndon et al., 2002*) such that older worms are physically able to move less well than reflected by their probability of dying. Another possibility might be related to the reduction in feeding worms exhibit with age (*Huang et al., 2004*). Feeding and locomotion are linked (*McCloskey et al., 2017*), so it might be that worms that have ceased feeding also tend to cease spontaneous locomotion.

## Scaling of behavioral decline during acute stress

It has been hypothesized that normal aging constitutes a low-level stress that results in the slow accumulation of damage leading to senescent decline. While the accelerated aging observed during oxidative stress in our experiments is far greater than what occurs during normal physiological processes, our result that the shape of functional decline is similar during normal aging and acute oxidative stress suggests a potential underlying similarity between these two conditions. Furthermore, we show that the relationship between individual decline and survival is conserved between oxidative stress and normal aging, indicating that the processes governing the unique shape of decline for short-lived and long-lived individuals are preserved. Our results indicate that the process of functional decline can be sped up by at least a factor of twenty while still maintaining a similar average shape. Together, our results suggest that normal aging and acute oxidative stress similarly impact the process of functional decline. Future work will aim to define mechanisms for this similarity.

If functional decline were dictated only by lifespan, we expect to observe a single curve relating the shape of decline to lifespan, regardless of environmental conditions. That is, interventions that drastically shorten lifespan, such as the addition of paraquat, should all exhibit an increase in decline rate between early and late life to be continuous with the curve presented in *Figure 6e*. Instead, however, we observe a translation of the tradeoff we observe during normal aging between the shape of functional decline and lifespan. This indicates that the shape of decline is not dictated by lifespan per se, but instead by the distance of an individual's lifespan relative to some standard in a given set of environmental conditions. For example, during normal aging, a lifespan of about 12 days results in neutral decline, or no change in decline rate between early and late life (*Figure 6a,d, e*), whereas during oxidative stress, a lifespan of about 12 hr results in neutral decline (*Figure 7i*). Together, these results suggest that relative to a standard lifespan in a given environment, there exists a defined shape of aging in the individuals whose lifespans differ from that standard as a result of genetic and/or stochastic factors.

## ODR-10 is required for elevated response threshold but not increased lifespan of *daf-2* mutants

We found that ODR-10 did not affect baseline locomotion of *daf-2* mutants, but did reduce the locomotory response of *daf-2* worms to an aversive stimulus. Furthermore, ODR-10 knockdown did not reduce *daf-2* lifespan, suggesting that elevated sensory response threshold is not required for increased lifespan of *daf-2* animals.

We found that ODR-10 suppressed locomotion in *daf-2* mutants only in response to stimulation, whereas Hahm *et al.* (*Hahm et al., 2015*) found that ODR-10 suppressed locomotion per se in *daf-2* mutants, One simple explanation for this discrepancy is that in Hahm *et al.*, locomotion assays were conducted soon after worms were manually stimulated due to picking onto the assay plate. The WorMotel allowed us to monitor worm behavior in a long-term unstimulated state in addition to after blue light stimulation. Future experiments may shed further light on whether reduced movement is required for the extended longevity of *daf-2*. The WorMotel will be useful in quantifying

behaviors that unfold over long periods of time and further exploring the relationships between behavior and lifespan.

## Additional axes of variation govern the aging process

It has been reported that survival curves scale by a multiplicative constant across a diverse set of genetic and environmental perturbations (*Zhang et al., 2016*). This result was interpreted as being compatible with the sum of biological inputs being filtered through a single state variable that determines the rate of aging. Our results indicate that a single variable is unlikely to be able to account for the inter-individual variability observed in aging. We observe variability in the shape of aging between individuals (*Figures 5a–f* and *6a–c*), indicating that the shape of aging does not scale for individuals with different lifespans. At the same time, we do observe a scaling of the relationship between the shape of aging and lifespan when the mean population rate of aging is accelerated with oxidative stress. Furthermore, we observe a difference in variability in aging decline across genotypes with differences in lifespan (*Figure 6g*). Therefore, our results indicate the likely existence of at least one additional variable across which the process of aging may vary between individuals within and across populations. Future work may uncover the full space across which the aging process may vary and mechanisms underlying variability in aging.

# Materials and methods

## Strains

The following strains were used in this study: N2, CF1038: *daf-16(mu86) I*, CB1370: *daf-2(e1370) III*, TJ1052: *age-1(hx546) II*, PR678: *tax-4(p678) III*, DA509: *unc-31(e928) IV*, KG1180: *lite-1(ce314) X*, RB754: *aak-2(ok524) X*, CX32: *odr-10(ky32) X*. All strains were maintained at 15°C under standard conditions (*Stiernagle, 2006*). All experiments were carried out at 20°C unless otherwise stated.

## WorMotel design and fabrication

To fabricate the WorMotel, we developed a 3D-printing based molding method (*Shepherd et al., 2011*). We designed a chip containing a rectangular array of either 48 or 240 rounded wells with 3 mm diameter, 3 mm depth, and center-to-center spacing of 4.5 mm (*Figure 1*). Each well was surrounded by a 0.5 mm wide and 3 mm deep channel, which would serve as the moat. Designs of the WorMotel masters were created using MATLAB. We printed a master corresponding to the negative of this shape with an Objet30 photopolymer 3D printer using the material VeroBlack. To mold the WM devices, we mixed Dow Corning Sylgard 184 PDMS according to the manufacturer's instructions and poured 35 g or 5 g of PDMS into the 240-well or 48-well masters, respectively. We then degassed the poured PDMS in a vacuum chamber for 1 hr or until no more bubbles were visible. Devices were cured overnight at 40°C and then removed from molds using a spatula.

## WorMotel preparation

To prepare devices for experiments, the chips were first treated with oxygen plasma for 4 min using a plasma cleaner (PE-50, Plasma Etch Inc., Carson City, NV or Plasmatic Systems Plasma Preen II, Plasmatic Systems. Inc., North Brunswick, NJ). This treatment renders PDMS temporarily hydrophilic, which greatly facilitates the filling of wells and moats. The medium was based on standard NGM media (*Stiernagle, 2006*) except low-gelling temperature agarose (gelling temp. 26–30°C, Research Products International, Mount Prospect, IL) was substituted for agar to minimize solidification of the agar during filling of the wells, and streptomycin (200 ng/mL) was added to the media to minimize bacterial contamination.

For lifespan experiments, but not for development experiments, we added 5-fluoro-2'-deoxyuridine (FUdR) to prevent growth of progeny. A frozen FUdR stock solution of 10 mg/ml in water was thawed and added to molten agar at 40°C at a concentration of 5 µL per mL just prior to filling. This yielded a final FUdR concentration of 200 µM. The moat solution consisted of 100 mM copper sulfate, which was approximately in osmotic equilibrium with the agar medium via the humidified air inside the chamber. The moat solution was added using a P200 pipette. About 15 µl of molten NGM agarose was added to each well. About 5 µl of a suspension of the *Escherichiae coli* bacterial strain DA837 (*Davis et al., 1995*), which is a streptomycin-resistant derivative of OP50

(*Brenner, 1974*) was added to each well after agarose solidification. For aging experiments, late-L4 worms were added to the WorMotel manually with a platinum wire pick.

PDMS devices were placed inside either a 90 mm petri plate for 48-well WorMotels or an Omni-Tray (Nunc Thermo Scientific) for 240-well WorMotels. 240-well WorMotels contained alignment tabs to keep devices in alignment with respect to the OmniTray. To maintain humidity inside the dishes, we used water-absorbing polyacrylamide crystals (AgSAP S, M2 Polymer, West Dundee, IL). Sterile water was added to the crystals in a ratio of 150:1 (water:crystals) by weight. Approximately 15 g of hydrated crystals were added around the WorMotel. We placed lids on all dishes. To prevent accumulation of water condensation, lids were prepared by coating with a 30% solution of Tween 20 (Sigma-Aldrich, St. Louis, MO) in water, which was allowed to dry before use. We wrapped Parafilm (Bemis, Nennah, WI) around the sides of the plate to reduce water loss while allowing sufficient gas exchange.

## Image acquisition

Images were captured with an Imaging Source DMK 23GP031 camera (2592 × 1944 pixels) equipped with a Fujinon lens (HF12.5SA-1, 1:1.4/12.5 mm, Fujifilm Corp., Japan). We used IC Capture (Imaging Source) or Phenocapture imaging software (http://phenocapture.com/) to acquire time lapse images through a gigabit Ethernet connection. For daily imaging we used the time schedule option in Phenocapture to record images every 5 s for a 30 min period twice daily. All experiments were carried out under dark-field illumination using four 4.7" red LED strips (Oznium, Pagosa Springs, CO) positioned approximately 2" below the WorMotel. Images were saved and processed by a 64-bit computer with a 3.40 GHz Intel Core i3 processor and 4 GB of RAM. Images were analyzed using custom-written MATLAB software.

Different spatial resolutions can be attained by adjusting the camera's field of view and thus by modulating the number of wells viewed at once. Imaging six wells at once yields approximately 5 μm resolution, imaging 12 wells yields 7 μm resolution, imaging 48 wells yields 15 μm resolution, and imaging 240 wells yields 36 μm resolution.

## Image processing

Temporally adjacent images were subtracted and divided by the average pixel intensity between the two images to generate normalized maps of pixel value intensity change. Depending on the task, time intervals of either 5 or 60 s were used to generate difference images. We refer to such difference images as 5-s or 60-s activity, respectively (see Aging Behavior Quantification). A Gaussian smoothing filter with standard deviation of one pixel was applied to the resulting difference image in order to reduce image noise. A binary threshold of 0.25 was used to minimize image noise was then applied to the filtered intensity change image in order to score whether or not movement occurred at each pixel location. All pixels in which movement occurred were summed up and the resulting value was called the 'activity' between the two frames.

## Blue light stimulation, uniform illumination, and temperature monitoring

To supply the blue light illumination, we use two high power LEDs (Luminus PT-121, Sunnyvale, CA) secured to an aluminum heat sink and connected in series. We used a relay (Schneider Electric, France) controlled by MATLAB through a LabJack (LabJack Corp., Lakewood, CO) or NIDaq (National Instruments, Austin, TX) interface to drive the LEDs at a current of 20 A through a power supply. To maximize the blue light irradiance and uniformity at the WorMotel, we constructed a box consisting of four acrylic mirrors with mirrored sides facing inwards and placed it around the WorMotel. We measured irradiance using a silicon power meter (Coherent, Santa Clara, CA). During aging experiments, blue light stimulation was applied once every twelve hours for 10 s. Temperature was continuously monitored with a temperature probe (LabJack EI1034) placed beside the sealed WorMotel.

A previous report (*Edwards et al., 2008*) found that continuous blue illumination at an irradiance of 2.8 mW/mm$^2$ kills N2 worms in 30 min. In our experiments, worms are subject to 20 s of blue light per day (10 s per stimulus, two stimuli per day). Therefore, it would take 90 days for worms to accrue 30 min of total illumination time with blue light. The irradiance of our blue light stimulus is

approximately five times weaker than that used in this Edwards *et al.* study. Assuming a linear response of blue light dosage toxicity, it would take 450 days for worms to accrue a toxic blue light dose in our experiments. Since the typical worm lifespan is between 15 and 30 days, we believe blue light toxicity in our experiments is not significant.

### Copper sulfate experiments

To test whether filling moats with a copper sulfate solution had any effects on worm development or survival, we prepared two WorMotels: in the first, moats were filled with NGM Buffer (NGMB), and in the second, moats were filled with 100 mM copper sulfate. NGMB consists of the same constituents as NGM agar (*Stiernagle, 2006*) but without peptone, cholesterol, or agar.

We manually added L3 larvae to each device and monitored the duration of the L4 stage as previously described (*Nelson et al., 2013*). When these worms reached the first day of adulthood, each animal was manually transferred to a new WorMotel in which the agar contained FUdR in order to assess the effect of moat solution on survival. Worms grown as larvae in the presence of a copper sulfate moat were transferred to a WorMotel in which moats were filled with copper sulfate; likewise for worms grown in the presence of an NGM moat. Lifespan was scored daily by manual methods (see below).

### RNAi by feeding

We used RNAi clones for EV, *daf-2*, and *odr-10* supplied by the Ahringer RNAi Libray (Source Bioscience, Nottingham, UK ). The bacterium was *E. coli* strain HT115. We induced RNAi in liquid culture for two hours using 1 µM IPTG. We added 2 µM IPTG and 25 µg/mL carbenicillin to molten WorMotel agar. Liquid bacteria suspensions were added on top of solidified agar. Worms were manually added to each well with a platinum wire pick.

### Lifespan determination

For manual assays, immobile worms were prodded three times with a platinum wire pick. Those that failed to respond were scored as dead. For automated assays, the maximum activity value recorded during the fifteen minutes after each light stimulus was used to determine time of death. Time of death was defined as the time point after the last time point for which the maximum 60-s activity was nonzero.

Any worm whose activity was uniformly zero beginning on day 2 of adulthood was assumed to have left its well or was not added due to experimental error, and these wells were censored from further analysis. 15 out of 1230 worms (1.2%) were censored in this manner. *tax-4* mutants were found to escape their wells as young adults at a much higher rate than all other strains. Therefore, *tax-4* mutants found to be absent from their wells at the conclusion of each experiment were censored from analysis. 18 out of 90 worms were censored in this manner.

### Aging behavior quantification

For each 30-min imaging epoch, spontaneous and stimulated locomotion were calculated as the maximum 60-s activity before and after the blue light stimulus, respectively. Spontaneous and stimulated locomotion reflect the maximum movement of an individual before or after blue light stimulus, respectively. The response duration was calculated as the total time during which the 5-s activity was greater than zero after the light stimulus. Response duration reflects the total time spent moving after the stimulus. The response latency was calculated as the time elapsed between the blue light stimulus and the first non-zero 5-s activity. The response latency reflects the time required for an individual to respond to an aversive blue light stimulus. Animals that did not move at all during the 15 min following blue light stimulation were assigned a response latency of 900 s.

### Decline rate calculation

We first considered the stimulated activity of each individual as a fraction of life rather than chronological time. We defined early-life as 20–60% of an individual's life. We defined late-life as 60–100% of an individual's life. Decline rate was calculated to be the negative of the slope of the stimulated activity during either early or late-life. Slope was determined with a linear fit in Matlab. For each

individual, the change in decline rate was calculated as the late-life decline rate minus the early-life decline rate.

### Paraquat treatment, imaging protocol, and data processing

300 mM paraquat stock solutions were prepared fresh each day. L4 animals were added to a WorMotel as normal. On either day 1 or day 9 of adulthood, 2 µL of paraquat stock solution were added to each well and allowed to dry. The final concentration of paraquat per well was about 40 mM, assuming uniform distribution into the agar. After addition of paraquat to each well, excess liquid was allowed to dry (about two hours), and plates were imaged continuously at 0.2 frames per second. Blue light stimulation occurred once per hour for 10 s. Stimulated activity for each animal was determined as the maximum activity between each light stimulus. Lifespan was determined automatically as described above.

### Comparison of behavioral decline during aging and stress

For a given population, we calculated stimulated activity as a fraction of life during aging and stress. We then corrected the activity during stress by calculating the average difference between the two quantities. By correcting for activity amplitude we could then ask whether the shape of decline was similar during aging and stress regardless of any offset. Finally, we calculated the normalized difference between aging and stress behavior at each fractional time point of life. The average normalized difference was reported as the percentage difference between behavior during aging and stress.

### Day replicates

Day replicates were defined as WorMotels prepared independently on separate days. For lifespan experiments involving DA837 bacterial food (*Figures 3*, *5* and *6*), at least three day replicates were performed for each strain. For lifespan experiments involving RNAi by feeding (*Figure 8*), two day replicates were performed. For survival experiments in which Paraquat was added at day 1 of adulthood (*Figure 7*), four day replicates were performed. For survival experiments in which Paraquat was added at day 9 of adulthood (*Figure 7b,d,f,h*), two day replicates were performed.

### Experimental design and statistical methods

Differences in lifespan and survival distributions (*Tables 1–2*) were compared using a Wilcoxon rank sum test. Behavioral comparisons (*Figure 6d*, *Figure 8a–h*) were performed using a two-tailed t-test.

## Acknowledgements

We thank Todd Lamitina and Vivek Sharma for helpful discussions and technical assistance. Some strains were provided by the CGC, which is funded by NIH Office of Research Infrastructure Programs (P40 OD010440). CCY was supported by a University Scholars Award. CFY was supported by the Alfred P Sloan Foundation, Ellison Medical Foundation, European Commission, and the National Institutes of Health.

## Additional information

### Funding

| Funder | Grant reference number | Author |
| --- | --- | --- |
| National Institutes of Health | R01-NS-084835 | Matthew A Churgin<br>Christopher Fang-Yen |
| Ellison Medical Foundation | AG-NS-1109-13 | Sang-Kyu Jung<br>Xiangmei Chen |
| European Commission | 633589 | Christopher Fang-Yen |
| National Institutes of Health | R01-NS-088432 | David M Raizen<br>Christopher Fang-Yen |
| Alfred P. Sloan Foundation | BR2012-084 | Christopher Fang-Yen |

The funders had no role in study design, data collection and interpretation, or the decision to submit the work for publication.

### Author contributions

MAC, Conceptualization, Resources, Software, Validation, Investigation, Visualization, Methodology, Writing—original draft, Writing—review and editing; S-KJ, Resources, Software, Investigation, Methodology; C-CY, Software, Investigation, Methodology; XC, Validation, Investigation, Methodology; DMR, Conceptualization, Methodology, Writing—review and editing; CF-Y, Conceptualization, Resources, Software, Supervision, Funding acquisition, Validation, Investigation, Visualization, Methodology, Writing—original draft, Project administration, Writing—review and editing

### Author ORCIDs

Matthew A Churgin, http://orcid.org/0000-0003-2299-0124
David M Raizen, http://orcid.org/0000-0001-5935-0476
Christopher Fang-Yen, http://orcid.org/0000-0002-4568-3218

## Additional files

### Supplementary files

• Supplementary file 1. Detailed protocol for WorMotel preparation.

• Supplementary file 2. Stereolithography (STL) file of the 240-well WorMotel mold.

• Supplementary file 3. Stereolithography (STL) file of the 48-well WorMotel mold.

• Supplementary file 4. Parts lists, instructions, and MATLAB software for imaging system and blue light stimulation apparatus.

• Supplementary file 5. MATLAB software suite for image processing and data analysis, with instructions.

• Supplementary file 6. Additional MATLAB scripts for data representation.

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
