## [Decision Letter]

Thank you for submitting your article "Longitudinal imaging of *C. elegans* in a microfabricated device reveals variation in behavioral decline during aging" for consideration by *eLife*. Your article has been favorably evaluated by Eve Marder (Senior Editor) and three reviewers, one of whom, Piali Sengupta (Reviewer #1), is a member of our Board of Reviewing Editors. The following individual involved in review of your submission has agreed to reveal their identity: Coleen T Murphy (Reviewer #3).

The reviewers have discussed the reviews with one another and the Reviewing Editor has drafted this decision to help you prepare a revised submission.

Summary:

This is a very nice Tools and Resource paper describing a new automated method (WorMotel) for longitudinal imaging of *C. elegans* during aging. The goal is to automate these studies to not only increase throughput but to also monitor behaviors of animals as they age. In particular, a major advantage of this method is the ability to follow the behaviors of individual animals over time as they age. As required for a Tools/Resource paper, the work describes the technology, documents basic controls that validate the approach, defines locomotory parameters of spontaneous and provoked movement, and makes accessible the WormMotel analysis software. A major strength is that the work addresses the variation in aging populations, distinguishing lifespan and "aging" indicators in a novel way. For example, the WorMotel is used to track individual life trajectories, allowing the authors to conclude that long and short-lived animals of different genotypes are in fact very similar to one another, but do not temporally scale as previously postulated. The WorMotel will be a valuable resource for the worm community.

Essential revisions:

1) Subsection “Mutants with short and long lifespan display patterns of late-life behavioral decline resembling those of short and long lived worms from a wild-type population”, third paragraph, arguments regarding extended twilight vs. universal scaling. This is an interesting dichotomy and the authors rightly distinguish this as a question to be resolved in the field. I think the accessibility of the author's arguments might be enhanced by the inclusion of a figure that illustrates the alternative hypotheses and anticipated outcomes.

2) At the simplest level, all animals might have the potential to exhibit rapid decline and plateau but the short-lived strains do not have the opportunity to exhibit this extended twilight, as they die early. Consider the data in Figure 4; at day 20-daf-16 both dies and is at the lowest activity number at this point; at day 20, daf-2 and N2 appear to be initiating a plateau stage in activity. I think it is important that the authors note the absence of the later plateau stage for daf-16 and short-lived N2, and that they note that daf-2 resemble long lived N2 suggesting the absence of a scaled response. Still, an unresolved question is why the data look as they do. This ambiguity might be better taken into account in interpreting consequences for aging biology in the discussion.

3) Figure 4 and analysis of bottom and top quartiles. I think the authors might also compare all quartiles in supplemental figures, and/or feature the middle quartiles. What does the middle sector look like for the 25% middle for N2; and at the individual level? Do individual curves support a temporal cutoff of the transition for shorter-lived animals in the group?

4) Figure 5 and paraquat stress. Paraquat induces high oxidative stress that is likely non-physiological; indeed, treated animals die 20X faster than untreated. The comparison to natural aging raises the apples/oranges issue. That shapes of decline curves are similar but the mechanistic question of why is unclear – this point should be discussed clearly.

5) Figure 6 and *odr-10*. The goal to test whether upregulation of *odr-10* in part underlies the reduced activity of daf-2 animals on bacteria is a worthy one, especially given the recent Hahm et al. 2015 paper. This experiment also highlights the use of the WorMotel for deep phenotyping. However, given the known caveats of RNAi in the nervous system, and the availability of *odr-10* null alleles, it is suggested that these experiments be repeated using an *odr-10* genetic mutant.

6) *Lite-1*. The use of the *lite-1* mutant is a bit confusing. It seems *lite-1* should be a good control for the induced activity measure, as the response to blue light is altered, but Figure 3—figure supplement 1, and Figure 4—figure supplement 1, suggest that response latency and duration can still be measured, so some response is indicated – why? How has *lite-1* previously been related to lifespan? Authors should better explain rationale on this.

7) A particularly strong advantage of this work is the ability to look at individual variability. While the reader can perhaps extract this information from the heat maps, it would be nice if the authors can address whether the extent of individual variability is altered (decreased/increased) in specific genotypes across the lifespan.

---

## [Author Response]

*Essential revisions:*

*1) Subsection “Mutants with short and long lifespan display patterns of late-life behavioral decline resembling those of short and long lived worms from a wild-type population”, third paragraph, arguments regarding extended twilight vs. universal scaling. This is an interesting dichotomy and the authors rightly distinguish this as a question to be resolved in the field. I think the accessibility of the author's arguments might be enhanced by the inclusion of a figure that illustrates the alternative hypotheses and anticipated outcomes.*

We have followed this suggestion by adding Figure 4.

*2) At the simplest level, all animals might have the potential to exhibit rapid decline and plateau but the short-lived strains do not have the opportunity to exhibit this extended twilight, as they die early. Consider the data in Figure 4; at day 20-daf-16 both dies and is at the lowest activity number at this point; at day 20, daf-2 and N2 appear to be initiating a plateau stage in activity. I think it is important that the authors note the absence of the later plateau stage for daf-16 and short-lived N2, and that they note that daf-2 resemble long lived N2 suggesting the absence of a scaled response. Still, an unresolved question is why the data look as they do. This ambiguity might be better taken into account in interpreting consequences for aging biology in the discussion.*

We have added an extended discussion of these questions in the subsection “Extension of late-life decrepitude in long-lived worms”.

*3) Figure 4 and analysis of bottom and top quartiles. I think the authors might also compare all quartiles in supplemental figures, and/or feature the middle quartiles. What does the middle sector look like for the 25% middle for N2; and at the individual level? Do individual curves support a temporal cutoff of the transition for shorter-lived animals in the group?*

We have moved analysis of quartile differences to a new Figure 6 and added data for the middle quartiles to Figure 6. We note that in Figure 6 all quartiles are plotted for each strain against each quartile's lifespan. In both panels 6D and 6E, a continuous trend is observed in the change in decline rate from shortest-lived to longest-lived worms.

We have also added Figure 6 and Figure 6—figure supplement 1, which display individual decline rate changes versus lifespan for each strain. This data also indicates that there is a smooth transition in the shape of behavioral decline between short-lived and long- lived individuals within a strain as well as across strains. We have added additional description of these results in the eighth paragraph of the subsection “Mutants with short and long lifespan display patterns of late-life behavioral decline resembling those of short and long lived worms from a wild-type population”.

*4) Figure 5 and paraquat stress. Paraquat induces high oxidative stress that is likely non-physiological; indeed, treated animals die 20X faster than untreated. The comparison to natural aging raises the apples/oranges issue. That shapes of decline curves are similar but the mechanistic question of why is unclear – this point should be discussed clearly.*

We have expanded our Discussion in the first paragraph of the subsection “Scaling of behavioral decline during acute stress”, to more clearly describe our interpretation of our paraquat results and how they may be expanded upon in the future.

*5) Figure 6 and odr-10. The goal to test whether upregulation of odr-10 in part underlies the reduced activity of daf-2 animals on bacteria is a worthy one, especially given the recent Hahm et al. 2015 paper. This experiment also highlights the use of the WorMotel for deep phenotyping. However, given the known caveats of RNAi in the nervous system, and the availability of odr-10 null alleles, it is suggested that these experiments be repeated using an odr-10 genetic mutant.*

We have indeed tested *odr-10(ky32)* mutants, which have either a partial or complete loss of ODR-10 function (Sengupta, Chou, & Bargmann, Cell 1996). This data, presented in Figure 6—figure supplement 1 in the original manuscript submission, has been moved to Figure 8 in the revised manuscript.

Our results from experiments testing *odr-10* mutants with *daf-2* RNAi (Figure 8) are consistent with those testing *daf-2* mutants with *odr-10* RNAi (Figure 8).

*6) Lite-1. The use of the lite-1 mutant is a bit confusing. It seems lite-1 should be a good control for the induced activity measure, as the response to blue light is altered, but Figure 3—figure supplement 1, and Figure 4—figure supplement 1, suggest that response latency and duration can still be measured, so some response is indicated – why? How has lite-1 previously been related to lifespan? Authors should better explain rationale on this.*

We have added a discussion of this justification to the fourth paragraph of the subsection “Mutant strains display diverse behavioral profiles during aging”.

We are not aware of any other studies of lifespan in *lite-1* mutants. We assayed *lite-1* worms in order to determine whether these mutants, previously shown to have a reduced response to blue light, can be assayed by our blue light illumination system for measuring stimulated activity. As pointed out by the reviewer, we found little difference in the responses to blue light between N2 and *lite-1* mutants, possibly because our light stimulus is very strong. We chose to include the *lite-1* mutant data in our manuscript because it indicates that even mutants with deficits in blue light response can be assessed by our method.

7) A particularly strong advantage of this work is the ability to look at individual variability. While the reader can perhaps extract this information from the heat maps, it would be nice if the authors can address whether the extent of individual variability is altered (decreased/increased) in specific genotypes across the lifespan.

We have added data for individual change in decline rate plotted against lifespan in Figure 6 as well as in Figure 6—figure supplement 1. This data allows readers to observe the trends in individual intra-strain and inter-strain variability.

We also compared the variation in decline rate change across strains and added this data as Figure 6. From this analysis we conclude that longer-lived strains exhibit less variation in the shape of behavioral decline compared with shorter-lived strains. We have added discussions of these results in the last paragraph of the subsection “Mutants with short and long lifespan display patterns of late-life behavioral decline resembling those of short and long lived worms from a wild-type population” and subsection “Additional axes of variation govern the aging process”.